# Label-Free Quantitative Proteomic Analysis of Nitrogen Starvation in Arabidopsis Root Reveals New Aspects of H_2_S Signaling by Protein Persulfidation

**DOI:** 10.3390/antiox10040508

**Published:** 2021-03-24

**Authors:** Ana Jurado-Flores, Luis C. Romero, Cecilia Gotor

**Affiliations:** Instituto de Bioquímica Vegetal y Fotosíntesis, Consejo Superior de Investigaciones Científicas and Universidad de Sevilla, 41092 Sevilla, Spain; ana.jurado@ibvf.csic.es

**Keywords:** Arabidopsis, autophagy, cysteine, hydrogen sulfide, persulfidation, proteomic

## Abstract

Hydrogen sulfide (H_2_S)-mediated signaling pathways regulate many physiological and pathophysiological processes in mammalian and plant systems. The molecular mechanism by which hydrogen sulfide exerts its action involves the posttranslational modification of cysteine residues to form a persulfidated thiol motif. We developed a comparative and label-free quantitative proteomic analysis approach for the detection of endogenous persulfidated proteins in N-starved *Arabidopsis thaliana* roots by using the tag-switch method. In this work, we identified 5214 unique proteins from root tissue that were persulfidated, 1674 of which were quantitatively analyzed and found to show altered persulfidation levels in vivo under N deprivation. These proteins represented almost 13% of the entire annotated proteome in Arabidopsis. Bioinformatic analysis revealed that persulfidated proteins were involved in a wide range of biological functions, regulating important processes such as primary metabolism, plant responses to stresses, growth and development, RNA translation and protein degradation. Quantitative mass spectrometry analysis allowed us to obtain a comprehensive view of hydrogen sulfide signaling via changes in the persulfidation levels of key protein targets involved in ubiquitin-dependent protein degradation and autophagy, among others.

## 1. Introduction

Hydrogen sulfide (including neutral H_2_S and the anionic forms hydrosulfide, HS^−^, and sulfide, S^2−^) is a well-established gasotransmitter that acts as a signaling molecule in all living organisms in which it has been studied, including humans and mammals in general [1,2], plants [3,4] and bacteria [5,6]. H_2_S functions in many physiological and pathological diseases in humans, such as Parkinson’s, Huntington’s and Alzheimer’s neurodegenerative diseases [7]. In plant systems, the number of known physiological processes involving signaling and regulation by H_2_S has grown rapidly in the last few years, which include responses to stress processes such as metal, drought, salinity, hypoxia or developmental processes such as seed germination and root development [4,8,9].

Persulfidation of cysteine residues has been described as one of the molecular mechanisms by which H_2_S exerts its signaling function [10,11]. Persulfidation, also called S-sulfhydration, refers to the protein posttranslational modification in which the functional thiol (-SH) group of cysteine is replaced with persulfide (-SSH) via the action of sulfide or polysulfide molecules. Persulfide residues are more nucleophilic and acidic than the original thiols and affect protein function and localization [12,13]. Since the first description of persulfidation, the number of known proteins that can undergo persulfidation and the physiological processes shown to be regulated by this mechanism have increased rapidly [12,14,15,16]. Although persulfides are very reactive, the development of chemical-specific protocols to label persulfidated proteins, such as the tag-switch method, has allowed the identification of these proteins by mass spectrometry [17,18]. In plants, proteomic analysis has revealed that a significant number of proteins are susceptible to modification by persulfidation; therefore, a large number of metabolic and regulatory pathways are affected by this modification [15,19]. Two of the processes that have been studied in greater depth are abscisic acid (ABA)-regulated stomatal opening/closure and the self-eating intracellular degradation system of autophagy.

In guard cells, H_2_S enzymatically produced by L-CYSTEINE DESULFHYDRASE 1 (DES1) is required for ABA-dependent NO and H_2_O_2_ production and stomatal closure [20,21,22]. Recently, several key components of the ABA signaling pathway that regulate stomatal opening/closure have been described to be regulated by H_2_S-dependent posttranslational persulfidation, such as open stomata 1 (OST1)/SNF1-RELATED PROTEIN KINASE2.6 (SnRK2.6) [23], RESPIRATORY BURST OXIDASE HOMOLOG PROTEIN D (RBOHD) and DES1 itself [24]. In this process, ABA induces the cysteine desulfhydrase activity of DES1 to catalyze the release of H_2_S, which leads to the persulfidation of DES1 itself and sustainable H_2_S accumulation to drive the persulfidation of OST1 and RBOHD, increasing their kinase and NADPH oxidase activities, respectively.

The regulation of the autophagy process in plants by H_2_S has been widely studied through the characterization of the DES1 enzyme [25,26]. DES1 belongs to the pyridoxal 5’-phosphate-dependent O-acetylserine(thiol)lyase family in Arabidopsis and shows L-cysteine desulfhydrase activity, which catalyzes H_2_S production from cysteine in the cytosol [27]. The *des1* null mutant line shows elevated ATG8 accumulation and a constitutive autophagy phenotype that can be restored by exogenous addition of H_2_S. Furthermore, exogenous addition of H_2_S can reverse the autophagy induced by C- or N- deprivation in wild-type plants [28,29]. Recently, the role of H_2_S in ABA-induced autophagy has been studied through the regulation of the activity of the cysteine protease ATG4 by regulating the persulfidation level of its catalytic Cys^170^ residue [30]. ATG4 persulfidation results in the inhibition of the protease activity of the enzyme to process the C-terminal end of ATG8, which is essential for autophagosome formation. Therefore, H_2_S acts as a repressor of autophagy processes under basal growth conditions by repressing ATG4 activity.

In the present work, we identified new target proteins of persulfidation related to N deprivation-induced autophagy by using the tag-switch method for the enrichment of persulfidated proteins and mass spectrometry identification with label-free quantitation.

## 2. Materials and Methods

### 2.1. Plant Ma Terial and Growth Conditions

Arabidopsis seeds were sown on MS solid medium containing 0.8% (*w*/*v*) agar and synchronized at 4 °C for 4 d. The plates were incubated vertically in a growth chamber under a long photoperiod regime as described [31]. For exposure to nitrogen deprivation conditions, one-week-old seedlings were transferred to the nitrogen-deficient MS solid medium for an additional 4 d of growth [29].

### 2.2. Immunoblot Analysis

Plant root material (100 mg) was ground in liquid nitrogen with 100–400 μL of extraction buffer (100 mM Tris-HCl, pH 7.5, 400 mM sucrose, 1 mM EDTA, 10 mg mL^−1^ sodium deoxycholate, 0.1 mM phenylmethylsulfonyl fluoride, 10 mg mL^−1^ pepstatin A and 4% [*v*/*v*] protease inhibitor cocktail [Roche]) using a mortar and pestle and was centrifuged at 500× *g* for 10 min to obtain the supernatant fraction as described previously [28]. The total amount of protein was determined using the Bradford described method [32]. For immunoblot analyses, 20 μg of root protein extract was electrophoresed on a 15% acrylamide gel before being transferred to a polyvinylidene fluoride membrane (Bio-Rad, Hercules, CA, USA) according to the manufacturer’s instructions. Anti-Cr-ATG8 [28] and secondary antibodies were diluted 1:2000 and 1:50,000, respectively, in PBS containing 0.1% Tween 20 (Sigma-Aldrich, St. Louis, MO, USA) and 5% milk powder. The ECL Select Western blotting Detection Reaction (GE Healthcare, Chicago, IL, USA) was used to detect proteins and for a protein loading control, the membrane before immunodetection was stained with Ponceau S (Sigma) to detect all protein bands.

### 2.3. Amino Acid Determination by UPLC-MS/MS

Approximately 100 mg of frozen plant tissue was homogenized in 1.5-mL Eppendorf tubes (Eppendorf, Hamburg, Germany) for 2 min at maximum speed with a Retsch ball mill (MM400; Retsch, Verder Scientific, Haan, Germany). The metabolites were extracted from each aliquot in 0.4 mL of 0.1 N HCl and 0.1% formic acid with shaking for 30 min at 4 °C in an Eppendorf ThermoMixer C. Samples were centrifuged for 15 min at 4700× *g* at 4 °C.

UHPLC separation of the amino acid sample was performed using the ExionLC^™^ UPLC system (Sciex, Framingham, MA, USA) with a reversed-phase column (100 mm × 4.6 mm × 100 Å particle, Kinetex XB-C18). The mobile phases were 0.1% formic acid in H_2_O (Buffer A, HPLC/LCMS grade) and 0.1% formic acid in acetonitrile (Buffer B, HPLC/LC-MS grade). A 5 µL sample was loaded per injection, and the gradient, applied at a flow rate of 600 µL min^−1^, was as follows: 3 min 100% A, 3 min linear gradient from 100% A to 80% A, 2 min linear gradient from 80% A to 50% A, 1 min hold at 50% A, 1 min linear gradient from 50% to 100% A, and hold at 100% A to re-equilibrate the column for 4 min (14 Q3 76.0 Da, DP 40.0 V, CE 17.0 V min total run time).

Mass spectra were acquired using a QTRAP 6500+ triple quadrupole (Sciex, Framingham, MA, USA) equipped with an electrospray ionization source operating in the positive ionization mode using an ion spray voltage of 4500 V. The other ESI parameters were as follows: curtain gas, 35 psi; collision gas, medium; temperature, 500 °C; nebulizer gas (GS1), 60 psi; and heater gas (GS2), 60 psi. Data were acquired with Analyst^®^ 1.7 software in the multiple reaction monitoring (MRM) mode with a detection window of 60 s. The ionization adducts measured [M+H^+^] and optimized declustering potential (DP) and collision energy (CE) for each MRM transition were Cys [Q1 122.0 Da]; Met [Q1 150.1 Da, Q3 104.0 Da, DP 6.0 V, CE 15.0 V]; Gly [Q1 76.0 Da, Q3 30.0 Da, DP 6.0 V, CE 19.0 V]; Ala [Q1 90.0 Da, Q3 44.0 Da, DP 6.0 V, CE 17.0 V]; Ser [Q1 106.0 Da, Q3 60.0 Da, DP 6.0 V, CE 15.0 V]; Pro [Q1 116.0 Da, Q3 70.0 Da, DP 20.0 V, CE 21.0 V]; Val [Q1 118.0 Da, Q3 55.0 Da, DP 11.0 V, CE 27.0 V]; Thr [Q1 120.0 Da, Q3 103.0 Da, DP 105.0 V, CE 25.0 V]; Ile [Q1 132.0 Da, Q3 86.0 Da, DP 8.0 V, CE 13.0 V]; Leu [Q1 132.0 Da, Q3 86.0 Da, DP 8.0 V, CE 1.0 V]; Asp [Q1 134.0 Da, Q3 74.0 Da, DP 7.0 V, CE 19.0 V]; Lys [Q1 147.0 Da, Q3 84.0 Da, DP 15.0 V, CE 23.0 V]; Glu [Q1 148.0 Da, Q3 84.0 Da, DP 21.0 V, CE 21.0 V]; His [Q1 156.0 Da, Q3 110.0 Da, DP 16.0 V, CE 19.0 V]; Phe [Q1 166.0 Da, Q3 103.0 Da, DP 11.0 V, CE 37.0 V]; Arg [Q1 175.0 Da, Q3 70.0 Da, DP 40.0 V, CE 27.0 V]; Tyr [Q1 182.0 Da, Q3 165.0 Da, DP 20.0 V, CE 13.0 V]; Gln [Q1 146.9 Da, Q3 84.1 Da, DP 16.0 V, CE 23.0 V]; Trp [Q1 204.9 Da, Q3 145.9 Da, DP 6.0 V, CE 23.0 V]; Asn [Q1 132.9 Da, Q3 86.9 Da, DP 6.0 V, CE 13.0 V]. Data were processed with Sciex OS^®^ software for peak integration and quantification.

### 2.4. Protein Persulfidation Enrichment by Tag-Switch Method

Plant root material (200 mg) was ground in liquid nitrogen with 400 µL of extraction buffer (50 mM Tris-HCl, pH 8.0, 1 mM EDTA, 1 mM phenylmethylsulfonyl fluoride and 4% [*v*/*v*] protease inhibitor cocktail [Roche]) using a mortar and pestle and was centrifuged at 500× *g* for 10 min to obtain the supernatant.

The tag-switch method was performed in 1 mg of protein extract as previously described [15]. The magnetic beads were removed, and the eluted proteins precipitated with TCA/acetone for mass spectrometry analysis and stored at −80 °C.

### 2.5. LC-MS/MS

The protein pellet obtained after the tag-switch labeling method was solubilized in digestion buffer containing 2% *w*/*v* sodium deoxycholate (SDC) in 100 mM Tris-HCl, pH 8.5, with 5 mM tris(2-carboxyethyl)phosphine (TCEP) and 30 mM 2-chloroacetamide (CAA). Trypsin and Lys-C were added at ratios of 1:50 and 1:100 (*w*/*w*), respectively. Digestion was performed overnight at 37 °C in a ThermoMixer C (Eppendorf, Germany) with shaking at 1000 rpm. SDC was removed via acid precipitation with trifluoroacetic acid (TFA). TFA was added until the pH reached 2, and the peptides were desalted using an Oasis HLB plate, dried and stored at −80 °C.

A total of 1000 ng of the tryptic peptide mixture was analyzed using an UltiMate 3000 high-performance liquid chromatography system (Thermo Fisher Scientific, Waltham, MA, USA) coupled online to a Q Exactive HF-x mass spectrometer (Thermo Fisher Scientific) via a nanoelectrospray source. Chromatography and tandem mass spectrometry conditions were as described [33].

### 2.6. Raw Data Processing and Analysis

All the raw files were analyzed by MaxQuant v1.6.17 software using the integrated Andromeda Search engine and were searched against the *Arabidopsis thaliana* UniProt Reference Proteome without isoforms (July 2020 release with 39,284 protein sequences). MaxQuant was used with the standard parameters (the “Label-Free Quantification” and “Match between runs” were selected with automatic values) except for the modifications: carbamidomethyl (C) was set as a variable modification together with oxidation (M), acetylation (Protein N-term), deamidation (N) and the other 4 that were manually set. In particular, we created Ciano-Biotin (addition of C(15) H(22) N(4) O(4) S), MethylTio (addition of C H(2) S), Sulfide (addition of S), and MSBT (addition of C(7) H(3) N S), which are all active only on cysteine residues [34]. The LFQ intensities found in ‘proteingroups.txt’ were filtered for reverse and potential contaminants, and the data were then analyzed in Perseus [35].

The mass spectrometry proteomics data have been deposited in the ProteomeXchange Consortium [36] via the PRIDE partner repository with the data set identifier PXD024061.

## 3. Results

### 3.1. Identification and Quantitative Comparison of the Persulfidation Patterns between Nitrogen-Sufficient and Nitrogen-Deprivation Conditions

Recently, it has been reported that the action of H_2_S in the regulation of the ABA-induced autophagic process is governed by the persulfidation of the cysteine protease ATG4a [30]. To assess whether additional target proteins are involved in the regulation and signaling of nonselective autophagy mediated by H_2_S, we used a label-free quantitative (LFQ) approach combined with the tag-switch method to measure protein persulfidation in root samples under nitrogen deprivation, a condition that has been extensively shown to induce autophagy in plants [29,37,38].

Protein samples from four biological replicates (independent pools) of root tissues from seedlings grown in sufficient nitrogen MS media (plus nitrogen samples, PN) or in N-deprived MS media (minus nitrogen samples, MN) were isolated and subjected to the tag-switch procedure (Figure 1A). The proteins eluted from streptavidin beads were digested, and the peptide solutions analyzed by liquid chromatography tandem mass spectrometry. To confirm that N deprivation treatment induced the autophagy process, we determined the ATG8/ATG8-PE protein levels in total protein extracts of the MN and PN samples before tag-switch labeling by immunoblotting; as expected, we detected a significant increase in these proteins after N deprivation (Figure 1B). Nitrogen deprivation blocks the incorporation of nitrogen into amino acids, decreasing the concentration of amino acids related with N-assimilation, such as glutamine (Gln), asparagine (Asn) or aspartate (Asp) [39]; therefore, to check the nutritional status of the samples, we determined the concentration of these amino acids and verified that the levels of Glu, Gln, Asp and Asn were indeed significantly reduced in N-deprived root samples (Appendix A).

A total of 5152 and 5057 proteins were identified as susceptible to persulfidation in the plus nitrogen samples (PN, Appendix A) and in the minus nitrogen samples (MN, Appendix A), respectively. Among these proteins, 4995 were common in both PN and MN samples, 157 were only present in PN samples and 62 were only present in MN samples (Figure 1C). In total, 5214 different proteins were identified as susceptible to persulfidation in root tissue (Appendix A). The largest proteomic analysis of persulfidation in leaf tissue published to date [15] reported a total of 3147 proteins as susceptible to persulfidation in leaves, of which 2523 were present in both root and leaf tissues (Figure 1D). The difference in the number of proteins detected between root and leaf tissues was more closely related to the use of an extra high-resolution mass spectrometer in the case of root samples than to this tissue showing a higher level of persulfidation than leaf tissue.

Most of the persulfidated proteins identified in root tissues showed this posttranslational modification in both N-sufficient medium and in N-starved medium. We have previously observed that an alteration of the persulfidation level of some proteins can be induced, e.g., by treatment with the ABA hormone, which can result in changes in the function or enzymatic properties of these proteins, as shown for the ATG4 cysteine protease [30]. To identify specific target proteins that may be involved in H_2_S-dependent signaling during N deprivation responses, we compared the persulfidation levels of the identified proteins by label-free quantification.

The LFQ proteomic approach led to the identification and quantification of 1519 proteins that were differentially persulfidated in response to N deprivation compared to the sufficient nutrient control condition, with an FDR threshold of 0.05 (*p* < 0.05) (Figure 2A). The Pearson correlation coefficient values of all the replicates of each sample were >0.9, indicating a high degree of correlation among samples (Figure 2B).

For further functional analysis, we added 155 persulfidated proteins that were exclusively identified under sufficient conditions or in N-starved samples, for a total of 1674 proteins with differential levels of persulfidation (Appendix A). Among these proteins, a group of 565 persulfidated proteins were only present or more persulfidated under N-starved conditions (Appendix A), and 1109 were not present or showed reduced levels of persulfidation under N-starved conditions (Appendix A). We have to state that the quantification was a relative comparison of minus N to plus N (MN/PN), i.e., a lower level of persulfidation in minus N equals a higher level of persulfidation in plus N.

### 3.2. Protein Persulfidation Have Impact on the Regulation of Protein Degradation and Autophagy Process

To assess the relevance of persulfidation on plant processes, we determined the subcellular location of the identified persulfidated proteins in roots, and the most represented compartments were the cytosol (20%), chloroplast (14%), ribosome (10%), Golgi apparatus, endoplasmic reticulum (ER) and cell wall (9%) (Figure 3A; Appendix A).

The biological processes targeted by persulfidation were dissected by their assignment and Gene Ontology (GO) analysis using UniProt (Figure 3B). The processes most represented in the analysis belonged to cellular processes and metabolism, which might be biased by the fact that these processes include the most abundant proteins, which are preferentially detected by MS, as observed in other proteomic analyses [15,40]. The 5214 identified proteins were also analyzed based on their assigned functions and classified into 35 functional groups using the MapMan nomenclature developed for plant-specific pathways and processes [41] (Figure 3C). The most abundant set corresponded to the general PROTEIN group, which included 20% of the total identified proteins with 1060 elements involved in protein degradation (346 elements), protein synthesis (341 elements), protein posttranslational modification (146 elements), and protein targeting (117 elements). Among the proteins involved in degradation, most were related to the ubiquitin-dependent degradation pathway (168). Additionally, the protein degradation bin included proteins related to the autophagy process, several of which were key proteins (17) involved in this process, such as the serine/threonine kinase TARGET OF RAPAMYCIN (TOR), its effectors proteins REGULATORY-ASSOCIATED PROTEIN OF TOR 1 (RAPTOR 1) and LETHAL WITH SEC THIRTEEN PROTEIN 8 (LST8), and several autophagy-related (ATG) proteins, including ATG2, 3, 5, 7, 11, and 13 and the cysteine protease ATG4, which regulates autophagy by processing and deconjugating the ubiquitin-like protein ATG8 (Appendix A) (Figure 4). In addition, up to 58 proteins in this bin were also related to endocytosis and the formation of the phagophore, including several transporters and vacuolar sorting proteins (Appendix A).

MapMan classification of the differentially persulfidated proteins showed that the most abundant group corresponded to the general classification of PROTEIN with 309 elements (Figure 3C), of which 225 proteins showed reduced levels of persulfidation under N deprivation and only 88 showed increased levels of persulfidation (Appendix A). The protein subgroup with degradation function, which had 101 elements, included several proteins involved in ubiquitin- and autophagy-dependent protein degradation containing nine E3-RING/U-BOX domain ligases, such as RGLG1 and RGLG5, with 2.7- and 0.5-fold increased persulfidation, respectively. These ligases negatively regulate drought stress responses by promoting the degradation of ABA-related PP2C phosphatases and MAPKKK18 [42,43]. The proteins involved in degradation pathways also included two E3-HECT domain ubiquitin ligases and three E3-SCF ubiquitin ligases, for example, MORE AXILLARY BRANCHES 2/ORESARA 9 (MAX2/ORE9) protein, which has been implicated in the regulation of important developmental and hormone processes [44,45,46] (Figure 5). In addition, the E2-like autophagy-related ATG3 protein, which catalyzes the conjugation of ATG8 and phosphatidylethanolamine, showed a 1.9-fold increase in persulfidation levels under N deprivation (Figure 4 and Figure 5).

### 3.3. Plant Hormone Signaling Components Are Targets of Persulfidation

Previous proteomic analyses performed in leaf tissue under nonstressed conditions have revealed that several proteins involved in the ABA signaling pathway are susceptible to persulfidation [15]. The new proteomic analysis performed in the present study showed a considerable number of root proteins involved in hormone metabolism (up to 101 elements, Figure 3C; Appendix A), such as ABSCISIC-ALDEHYDE OXIDASE 2 and 3 (AAO2, AAO3, respectively). In addition, the analysis showed a significant number of proteins involved in hormone signaling (Appendix A) including several ABA signaling components, such as the receptors PYRABACTIN RESISTANCE 1 (PYR1) and PYR1-LIKE 1 and 2 (PYL1, PYL2); downstream elements, such as PROTEIN PHOSPHATASE 2C 7 (HAB2); five SNF-related serine/threonine kinases (SnRK2.1, 2.2, 2.3, 2.4, 2.10); and jasmonic acid and brassinosteroid signaling, such as JAR1, COI1, EIN2, BRI1 or BSK1 [47,48]. In addition to the serine/threonine-protein kinases mentioned above, we also identified up to 240 proteins involved in signaling processes, which included 66 protein kinases with important regulatory functions, such as the mitogen-activated protein kinases MPK3 and MPK6; MPKK4/5; the SNF1-related protein kinase catalytic subunits alpha KIN10 and KIN11, which have also been identified as regulators of autophagy [49,50]; and the receptor-like protein kinases FERONIA and THESEUS 1.

### 3.4. Cellular Processes and Branched-Chain Amino Acid Biosynthesis Are Regulated by Persulfidation under N Deprivation

The classification overview by cell function showed an overrepresentation of proteins that were less persulfidated under N deprivation in the categories of protein synthesis (106 out of 115 differentially persulfidated proteins), amino acid activation (11 out of 12), regulation of transcription (69 proteins out of 78), RNA processing (24 out of 32), DNA synthesis (16 proteins out of 16) and RNA synthesis (8 out of 8) (Figure 6)

The overrepresentation test analysis performed with Database for Annotation, Visualization and Integrated Discovery (DAVID) showed 42 categories within the GO biological process classification with more than two-fold enrichment (FDR < 0.05). The proteins with a higher level of enrichment corresponded to the categories involved in branched-chain amino acid biosynthesis (8.6-fold), with more than 13 proteins, as well as aspartate, valine, leucine and isoleucine biosynthesis (Appendix A). To reduce redundancy in the classification annotation, we performed functional annotation clustering to group similar annotations, and three groups with significant enrichment scores were generated (Table 1). The largest cluster, with five annotation processes, included branched-chain amino acid (BCAA) biosynthesis and the leucine, valine, isoleucine (BCAA amino acids) and aspartate amino acid biosynthesis pathways, which showed reduced levels of persulfidation in most of the enzymes involved in these pathways (Figure 7). Amino acid content analysis showed that under N deprivation, there were increases in valine, leucine and isoleucine concentrations, which suggests that a reduction in persulfidation may increase the enzyme activity of the BCAA pathway to redirect amino acid synthesis (Figure 7). A second cluster, with three annotation processes, included glycine and serine metabolism together with tetrahydrofolate interconversion. Finally, a third cluster comprised four annotations related to chitin and cell wall catabolism as well as polysaccharide and amino sugar metabolism. Within this cluster, the cell wall DUF642 protein, one of the proteins with the highest persulfidation change, with a more than 100-fold change in reduction under N deprivation, was included.

## 4. Discussion

The combination of the chemical selective tag-switch method to label and purify persulfidated proteins coupled to liquid chromatography-tandem mass spectrometry allowed us to identify 5214 proteins susceptible to persulfidation in Arabidopsis root samples. These proteins represented 13% of the entire annotated proteome of Arabidopsis, which includes 39,346 proteins (UniProt database) produced from 27,655 protein-coding genes [51]. Recently, a comprehensive and extensive proteomic study of 30 different Arabidopsis tissues using state-of-the-art mass spectrometry has reported a protein atlas that covers 18,210 proteins [52]; therefore, the 5214 persulfidated proteins detected in root tissue in the present study represent almost 30% of the technically detectable proteins by currently available mass spectrometry technology.

A previous proteomic study of persulfidation in Arabidopsis leaf tissue using the selective tag-switch method also reported the identification of 2015 persulfidated proteins, although a less-sensitive and lower-resolution mass spectrometry method was used [15]. The GO classification of biological functions of the identified persulfidated proteins in root and leaf tissue belonging to cellular and metabolic processes is biased by the fact that more abundant proteins are detected more favorably by MS with underrepresented protein evidence for low abundant transcripts [52,53]. This is a difficult problem to solve in high-throughput proteomic related with sensitivity due to the high dynamic range in abundance that range from millions of molecules for structural and housekeeping proteins to few ones per cell for transcription factor and signaling proteins [54]. In addition to primary and energy cellular metabolism, based on the assigned functions for plant-specific pathways and processes, almost one-fifth of the identified proteins were involved in cellular processes related to protein synthesis, ubiquitin-dependent degradation and autophagy and protein posttranslational modification. H_2_S/persulfidation has been widely linked to the autophagy process in plants by preventing ATG8 accumulation, acting as a repressor of autophagy under sufficient nutrient conditions [25,28,29]. Several proteins involved in the formation of the autophagosome core have been shown to be susceptible to persulfidation in leaves, such as ATG5, the E1-like protein ATG7 and the E2-like protein ATG3, which have essential cysteine residues for the formation of thioester intermediates [15]. In this work, we expanded the spectrum of persulfidated proteins involved not only in the formation of the autophagosome but also in the initial regulatory steps of autophagy, such as the negative regulator TOR kinase and its effectors RAPTOR and LST8, which are active under sufficient nutrient conditions to upregulate protein translation and suppress autophagy. Under nutrient deficiency (e.g., N deprivation used in this work), inactive TOR cannot phosphorylate ATG13 and allow ATG1 complex formation and the progression of autophagy. Other regulators upstream of TOR susceptible to persulfidation are the KIN10 and KIN11 catalytic subunits of SnRK1, which appears to act as a positive regulator upon nutrient starvation [49].

In the final steps of autophagosome formation, the cysteine protease ATG4 cleaves the C-terminal portion of ATG8 at a conserved Gly residue that is initially conjugated to the catalytic Cys of ATG7 to form a thioester bond ATG7~ATG8 intermediate. After a transthiolation reaction, ATG8 is sequentially transferred to the E2-like enzyme ATG3, generating a new thioester ATG3~ATG8 conjugate that is finally transferred from the Cys residue of ATG3 to phosphatidylethanolamine (PE) to be incorporated in the phagophore [55]. Quantitative analysis of the persulfidation level under N deprivation compared to that under sufficient nutrient conditions allowed us to observe only significant changes in the level of persulfidation of the conjugating enzyme ATG3. Although we did not investigate the effect of persulfidation on the ATG3 reaction mechanism, considering that persulfides are more nucleophilic and acidic than the corresponding thiol, we expect that persulfidation increases the affinity and transthiolation reaction between ATG3 and ATG7 [56,57]. This increased reactivity can shift the process, leading to increased ATG8-PE formation and autophagosome formation, as observed under N deprivation (this work) [29]. This is in contrast to recent studies on the regulation of H_2_S/persulfidation of ABA-dependent induced autophagy, in which hormone treatment induces a transient decrease in the level of persulfidation of the protease ATG4, leading to 5.2-fold less persulfidation after 3 h, resulting in activation of the processing activity of the protein to cleave the C-terminal portion of ATG8 to release the Gly residue for PE conjugation [30]. Although the nucleophilic nature of Cys is also a factor in the catalytic mechanism of cysteine proteases, molecular modeling shows that persulfidation of Cys^170^ of ATG4 causes conformational changes and intramolecular rearrangements of the catalytic site, which affect ATG8 substrate recognition [30]. Thus, the H_2_S-dependent molecular mechanism that regulates autophagy can be multiple and dependent on the physiological process that triggers it.

In addition to autophagy-related proteins, quantitative analysis also showed significant changes in the persulfidation level of ubiquitin-dependent proteasome degradation proteins. The most relevant and abundant group of proteins corresponded to several RING/Ubox and HECT domain E3 ubiquitin ligases with increased levels of persulfidation and three SCF domain E3 ligases with reduced levels of persulfidation when plants were subjected to N deprivation conditions. RING domain E3 ligases comprise a family with a domain signature of 46–60 residues, such as Cys-X_2_-Cys-X(_9–39_)-Cys-X(_1–3_)-His-X(_2–3_)-Cys/His-X_2_- Cys-X(_4–48_)-Cys-X_2_-Cys, where Cys and His are metal ligands that bind two zinc atoms and X is any amino acid [58]. E3 ubiquitin ligases act by bringing the targeted protein substrate close to thioesterified Ub~E2 so that ubiquitin is transferred from E2 to the substrate. Two of the identified persulfidated RING domain ligases are the RGLG1 and RGLG5 E3 proteins, which have been described to interact with the protein phosphatase PP2CA after ABA promotion to mediate ubiquitination and protein degradation of phosphatase [59]. In addition, RGLG1 has also been shown to regulate the protein level of PROTEIN KINASE KINASE KINASE 18 (MAPKKK18), which mediates the MAPK cascade and plays important roles in senescence, ABA signaling, and drought tolerance [43]. Both RGLG1 and RGLG5 contain a unique RING domain at the C-terminal end of the protein and two or four internal Cys residues, respectively. The coordination of cysteine residues to zinc in the RING domain protects them from oxidation, but internal Cys residues can be targets for persulfidation. This is the case in the E3 RING ligase PARKIN, where nitrosylation decreases its ubiquitination capacity but persulfidation increases its ubiquitination capacity [60]. Therefore, with these considerations, it can be hypothesized that the increase in persulfidation of RGLG1-5 E3 ligase proteins results in an increase in ubiquitination activity, leading to the transfer of Ub from E2 to the substrate and therefore its degradation by the proteasome.

Another component of the SCF (ASK-cullin-F-box) E3 ligase protein identified in this work is MAX2/ORE9, which reduces its persulfidation level by 30-fold via N deprivation. MAX2 belongs to the F-box leucine-rich protein family and is a central component in the strigolactone and karrikin signaling cascade [61]. MAX2 contains 24 Cys residues within its sequence, many of which are part of LRR domains; therefore, persulfidation of one or several of those Cys residues may result in conformational changes of the α/β horseshoe fold of the leucine-rich repeat.

Nitrogen nutrition is an essential factor for plant growth and productivity, as its deficiency results in a reduction in chlorophyll and Rubisco enzymatic activity and a marked decrease in photosynthesis [62,63]. Quantitative proteomic analysis of N-starved Arabidopsis seedlings revels altered abundances of 29 photosynthesis-related proteins functioning as components of the photosynthetic I and II complexes, the electron transport machinery and enzymes in the Calvin cycle, indicating reduced CO_2_ assimilation rates [64]. Moreover, the regulatory interaction between assimilatory nitrate and sulfate reduction is well established [65,66], so this proteomic analysis also finds significant alterations of relevant enzymes of the sulfur assimilation pathway, such as reduced levels of chloroplast sulfite reductase (SIR) and O-acetylserine(thiol)lyase B, which sequentially catalyze the reduction of sulfite to sulfide and its incorporation into O-acetylserine to form cysteine [67]. The level of cytosolic O-acetylserine(thiol)lyase A is also reduced under N deprivation, and it upregulates the level of mitochondrial ß-cyanoalanine synthase C1, which produces sulfide as a reaction product [64,68]. Therefore, alterations of the levels of these critical enzymes may result in specific adjustments of the level of H_2_S in the three compartments by reducing its level in the chloroplast and increasing its level in the cytosol and mitochondria [4]. We have observed that when faced with a nutritional or hormonal stimulus, the protein persulfidation pattern does not change in a single direction, increasing or decreasing, but rather changes occur in both directions (this work) [30]. This effect may be caused by temporal and localized (subcellular) concentration changes in the sulfurating agent that refine the stimulus response through changes in the persulfidation levels of target proteins, as observed for ATG4 in the ABA-induced autophagy response [30] and DES1, RBOHD and OST1 in ABA stomatal opening/closure [23,24].

## 5. Conclusions

The proteomic analysis performed in this work provides a list of proteins that are likely to be modified by cysteine persulfidation in plants and deepens the understanding of the regulatory role of hydrogen sulfide in essential processes in plants, such as autophagy, selective ubiquitin-dependent degradation of proteins and ABA hormonal regulation. Quantitative analysis of the protein persulfidation level under N deprivation indicates new protein targets involved in H_2_S-mediated intracellular signaling and metabolic readjustment that will have to be studied in greater depth.

## Figures and Tables

**Figure 1 antioxidants-10-00508-f001:**
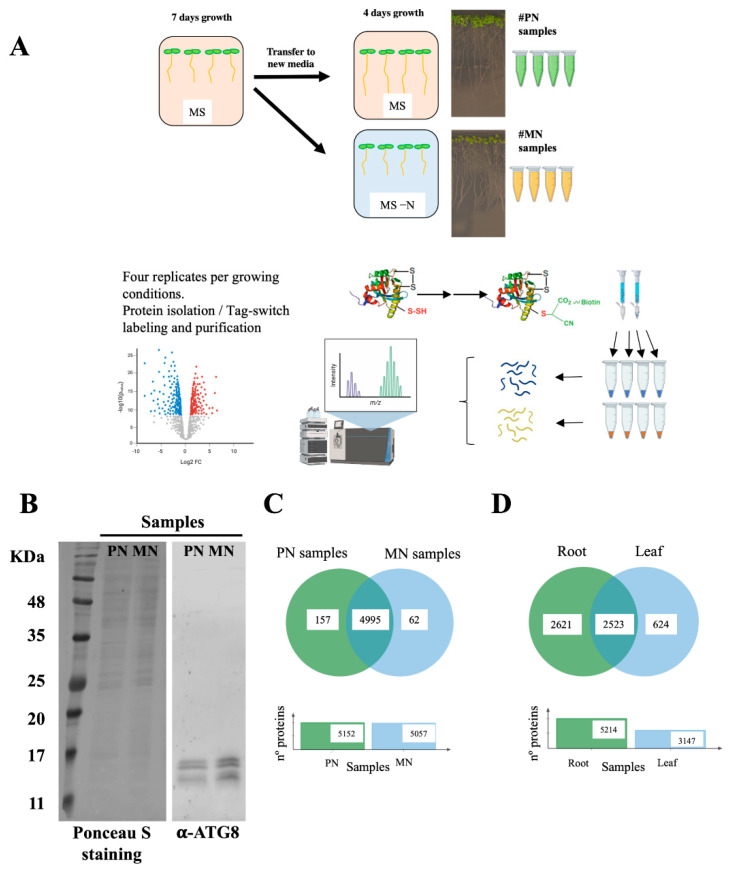
Proteomic analysis of protein persulfidation in response to N starvation in root tissue. (**A**) Workflow of the root sample preparation followed by tag-switch protein labeling with CN-biotin, protein purification, tryptic digestion and quantitative DIA analysis of eluted proteins. (**B**) Immunoblot analysis of ATG8 protein accumulation in roots. Total protein extracts from the PN and MN samples were subjected to immunoblot analysis with anti-ATG8. As a protein control, Ponceau S staining of the blot is shown. (**C**) Venn diagram showing the intersection of the persulfidated protein sets identified in the root samples grown in N-sufficient media (PN samples) and N-depleted media (MN samples). (**D**) Venn diagram showing the intersection of the persulfidated protein sets identified in root tissue and reported leaf tissue [15].

**Figure 2 antioxidants-10-00508-f002:**
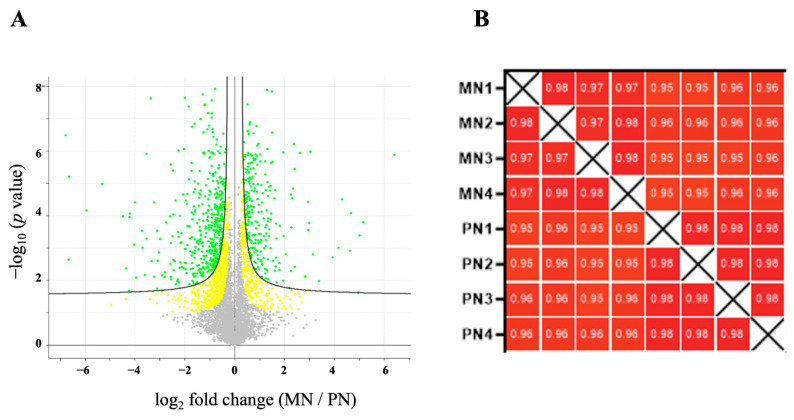
Quantification of the persulfidated proteins identified in root tissue grown in N-starved media compared to N-sufficient nutrient media. (**A**) Volcano plot illustrating significantly differentially abundant proteins. The −log_10_ (*p* value) is plotted as the log_2_ of the fold change between N-starved samples (MN) and N-sufficient samples (PN). Red dots represent protein with FDR < 0.01 and yellow dots FDR < 0.05. (**B**) Pearson correlation coefficients of the four independent N-starved (MN) and N-sufficient nutrient samples (PN).

**Figure 3 antioxidants-10-00508-f003:**
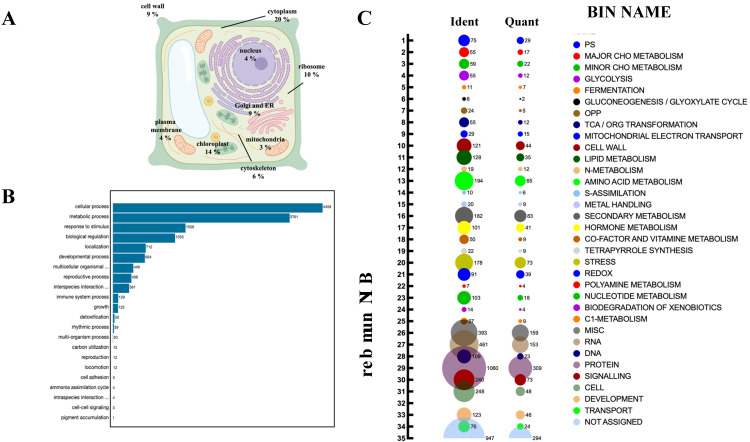
Subcellular localization and functional classification of the persulfidated proteins in root tissues. (**A**) Subcellular distribution of the persulfidated proteins classified by GO analysis (UniProt) as being related to cellular components. (**B**) GO classification (UniProt) of proteins related to biological processes. (**C**) Bubble plot of the functional classification of the identified (Ident) and quantified (Quant) proteins according to the plant-specific database MapMan.

**Figure 4 antioxidants-10-00508-f004:**
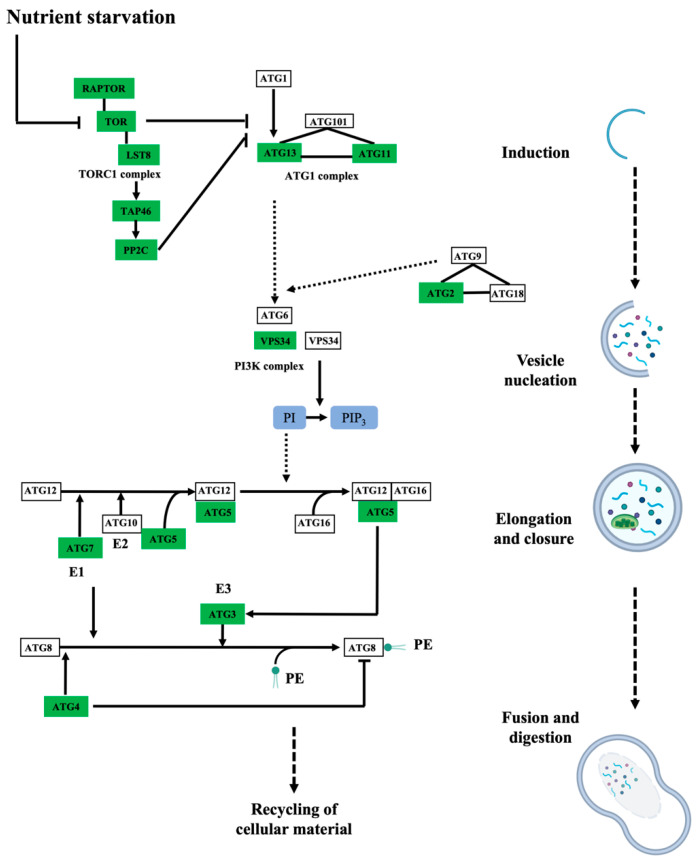
Persulfidated proteins identified in root tissues involved in autophagy. Simplified schematic drawing showing the persulfidated proteins identified as being involved in autophagy induction, vesicle nucleation, autophagosome elongation and closure, vacuole fusion and material digestion. The rectangular boxes represent the abbreviated name of the proteins involved in the autophagy pathway, and the green color of the boxes indicates proteins identified in this proteomic analysis. Adapted from the Kyoto Encyclopedia of Genes and Genomes (KEGG) Pathway database resource for pathway mapping.

**Figure 5 antioxidants-10-00508-f005:**
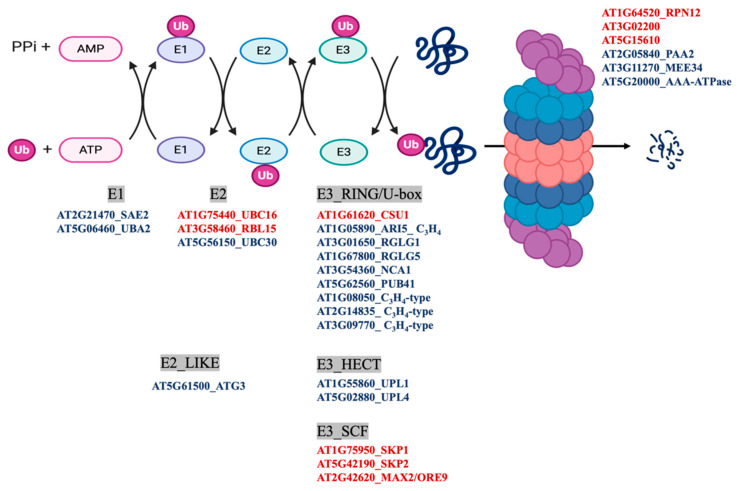
Ubiquitin- and autophagy-dependent degradation pathways. The scheme shows the differentially persulfidated proteins identified under N-starved growth conditions compared to sufficient N media. Red indicates proteins with reduced persulfidation in the N-starved sample, and blue indicates proteins with increased persulfidation levels.

**Figure 6 antioxidants-10-00508-f006:**
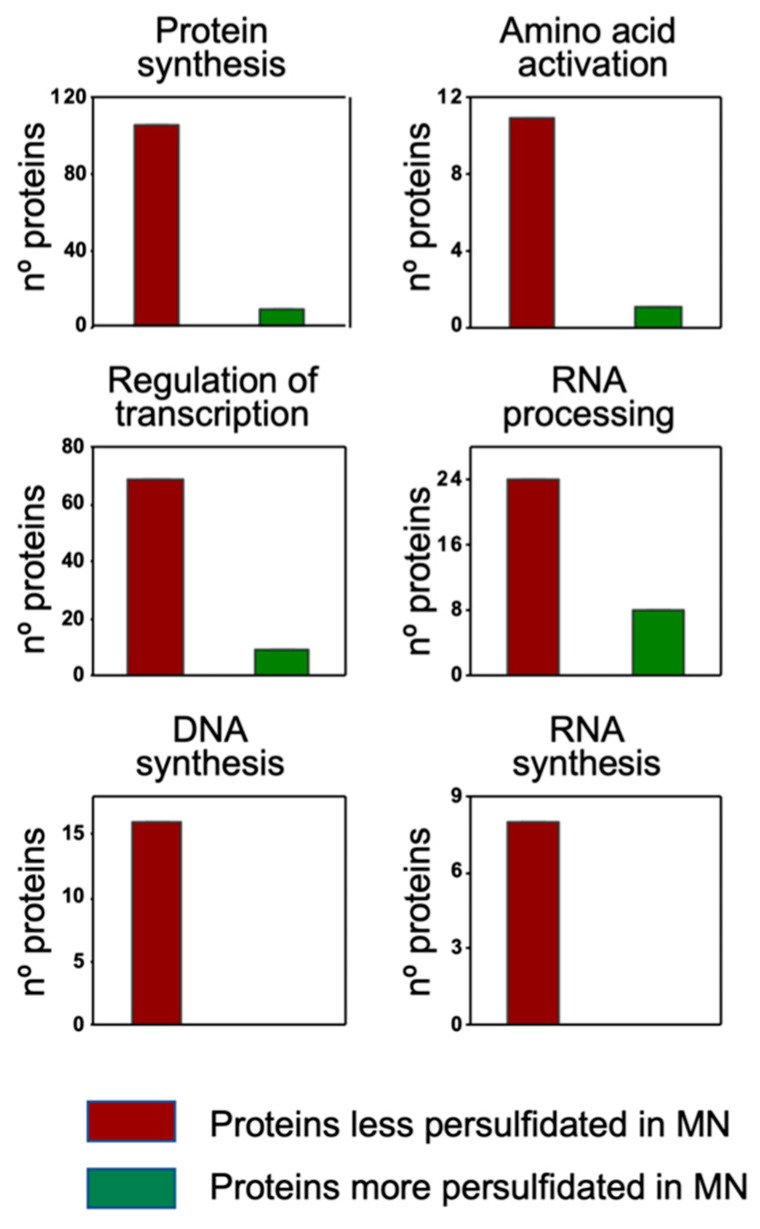
MapMan classification by Cell Function of the differentially persulfidated proteins in N-starvation (MN) medium compared to N-sufficient medium (PN). Adapted from the Cell_function overview_mapping image.

**Figure 7 antioxidants-10-00508-f007:**
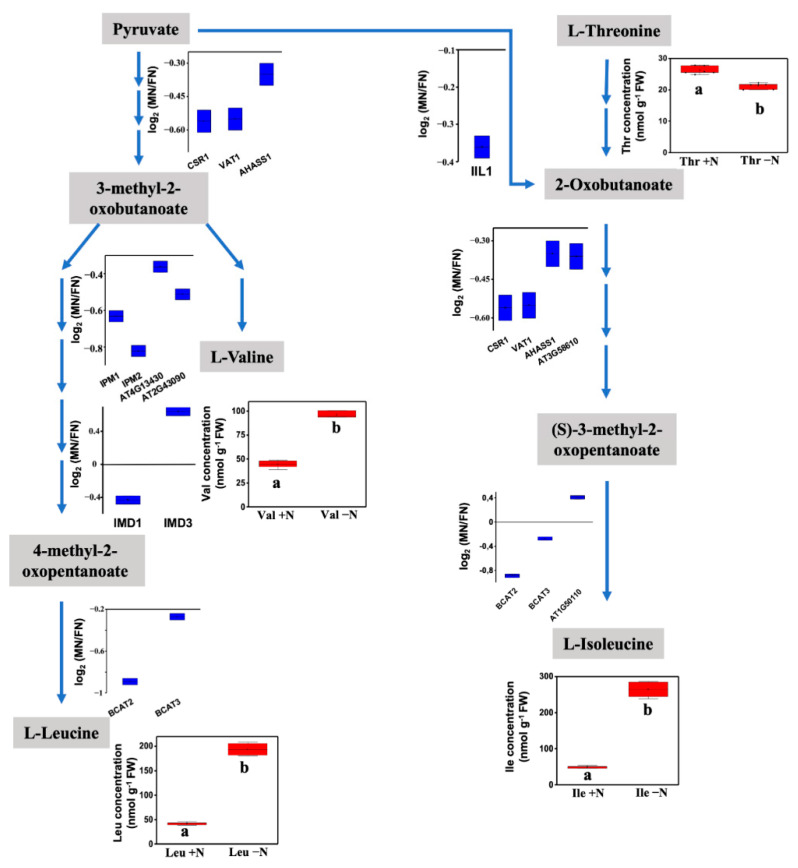
Schematic of branched-chain amino acid metabolism. The blue box plot shows the quantified (log_2_-fold change MN/PN) level of persulfidation of the enzymes involved in the process detected in N-starved samples compared with N-sufficient samples. The red box plot shows the amino acid concentration measured in N-deprived (-N) or N-sufficient samples (+N) (n = 4). Different letters below the bars indicate significant differences with *p* < 0.01 (One-way ANOVA, post-hoc Tukey adjustment).

**Table 1 antioxidants-10-00508-t001:** Functional annotation clustering of the differentially persulfidated proteins.

**Cluster 1. Enrichment Score: 5.8**					
**Term**	**Count**	***p* Value**	**Fold Enrichment**	**Benjamini**	**FDR**
GO:0009082~branched-chain amino acid biosynthetic process	13	2.44 × 10^−9^	8.57	3.63 × 10^−7^	3.52 × 10^−7^
GO:0009098~leucine biosynthetic process	13	1.08 × 10^−8^	7.79	1.32 × 10^−6^	1.28 × 10^−6^
GO:0009099~valine biosynthetic process	10	1.12 × 10^−6^	7.75	7.89 × 10^−5^	7.64 × 10^−5^
GO:0009097~isoleucine biosynthetic process	7	3.31 × 10^−4^	6.59	1.19 × 10^−2^	1.16 × 10^−2^
GO:0006532~aspartate biosynthetic process	6	4.53 × 10^−4^	7.91	1.49 × 10^−2^	1.44 × 10^−2^
**Cluster 2. Enrichment Score: 1.7**					
**Term**	**Count**	***p* Value**	**Fold Enrichment**	**Benjamini**	**FDR**
GO:0006544~glycine metabolic process	4	0.012055	7.53	2.31 × 10^−1^	2.24 × 10^−1^
GO:0035999~tetrahydrofolate interconversion	5	0.022806	4.39	3.73 × 10^−1^	3.61 × 10^−1^
GO:0006563~L-serine metabolic process	4	0.025800	5.86	4.02 × 10^−1^	3.89 × 10^−1^
**Cluster 3. Enrichment Score: 1.2**					
**Term**	**Count**	***p* Value**	**Fold Enrichment**	**Benjamini**	**FDR**
GO:0006032~chitin catabolic process	6	0.03689	3.16	4.75 × 10^−1^	4.61 × 10^−1^
GO:0016998~cell wall macromolecule catabolic process	6	0.04293	3.04	5.38 × 10^−1^	5.21 × 10^−1^
GO:0000272~polysaccharide catabolic process	5	0.05997	3.296	6.64 × 10^−1^	6.43 × 10^−1^
GO:0006040~amino sugar metabolic process	4	0.09970	3.51	9.50 × 10^−1^	9.21 × 10^−1^

## Data Availability

The mass spectrometry proteomics data have been deposited in the ProteomeXchange Consortium via the PRIDE partner repository with the data set identifier PXD024061.

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
