# Peer review of "Label-Free Quantitative Proteomic Analysis of Nitrogen Starvation in Arabidopsis Root Reveals New Aspects of H2S Signaling by Protein Persulfidation"

_antioxidants, 2021, doi:10.3390/antiox10040508_

Round 1

Reviewer 1 Report

The manuscript by Jurado-Flores et al. reports that label-free quantitative proteomic analysis of nitrogen starvation in Arabidopsis root reveals new aspects of H2S signaling by protein persulfidation. It contains a set of novel interesting data, while the presentation, in my opinion, could be improved significantly. Thus I think that there is still great potential for the improvement of the text organization within the results section starting from the general analysis of differentially persulfidated proteins finalizing by the identification of the particular pathways in ubiquitin-dependent protein degradation and autophagy (general overview by cell function follows protein degradation and autophagy pathways in the current version). I think that the logical sequence of results presentation would be very helpful for further reader’s understanding. Proper subheadings containing more from the results then from the methods would help to perform such reorganization.

Minor:

Lane 210: glutamine listed as glutamate (Glu)

Author Response

We have organized the results section according to the reviewer's comment and included new subheadings according to the results described instead of the methods. The new subheadings are:

  1. Results

3.1 Identification and quantitative comparison of the persulfidation patterns between nitrogen-sufficient and nitrogen-deprivation conditions

3.2 Protein persulfidation have impact on the regulation of protein degradation and autophagy process

3.3 Plant hormone signaling components are targets of persulfidation

3.4 Cellular processes and branched-chain amino acid biosynthesis are regulated by persulfidation under N deprivation

The abbreviation of glutamine has been corrected

Reviewer 2 Report

Dear authors, 

I have only a few minor comments on your manuscript which are provided in the attached file. I appreciate the novelty of your topic of protein persulfidation in response to nitrogen starvation. 

Final recommendation Accept after a minor revision. 

Author Response

Results, line 210: I have only one comment related to methods and results and nitrogen-containing amino acids. I think that using the term „nitrogen-containing amino acids“ for glutamate plus glutamine and aspartate plus asparagine only is very confusing since in fact, by definition, all amino acids contain nitrogen. Moreover, amino acids arginine and lysine also contain extra N except for the amino groups involved in peptide bonds. i thus would like to ask the authors why they did not focus also on arginine and lysine in addition to Asp+Asn and Glu+Gln. Moreover, the authors should not use the term „nitrogen-containing amino acids“ only for Asp+Asn and Glu+Gln since by definition, all amino acids contain nitrogen.

We have not attempted to do a thorough amino acid analysis. We only want to have some data confirming nutritional stress in addition to the phenotypic observation of root shortening. The term “nitrogen-containing amino acids“ has been removed.

Results, line 257: Correct the typing error in the enzyme name „protein phosphatase 2C7“ (NOT „protein phospahtase 2C7“).

Results, line 385: Add a dot at the end of the sentence „…that triggers it.“

Discussion, line 419: Correct the typing error in the verb „reveals“ (NOT „revels“).

The mistakes have been corrected

Results, line 345: The statement „more abundant proteins are detected more favorably by MS“ is interesting since as I know an analogous situation is true for 2-DE gels, i.e., that more abundant proteins such as RubisCO in leaf tissues, overlap (cover) several less abundant proteins with similar pI and MW values on 2-DE gels. This fact has often been cited as a major disadvantage of 2-DE approach in comparison to gel-free MS/MS approaches. I think that the authors should provide a brief explanation to their statement.

A short explanation has been added.

This is a difficult problem to solve in high-throughput proteomic related with sensitivity due to the high dynamic range in abundance that range from millions of molecules for structural and housekeeping proteins to few ones per cell for transcription factor and signaling proteins 

Wang, M.; Weiss, M.; Simonovic, M.; Haertinger, G.; Schrimpf, S.P.; Hengartner, M.O.; von Mering, C. PaxDb, a database of protein abundance averages across all three domains of life. Mol. Cell. Proteomics 2012, 11, 492-500,